# Comparison of the Direct Identification and Short-Term Incubation Methods for Positive Blood Cultures via MALDI-TOF Mass Spectrometry

**DOI:** 10.3390/diagnostics14151611

**Published:** 2024-07-26

**Authors:** Shu-Fang Kuo, Tsung-Yu Huang, Chih-Yi Lee, Chen-Hsiang Lee

**Affiliations:** 1Department of Laboratory Medicine, Chiayi Chang Gung Memorial Hospital, Chiayi 613, Taiwan; ivykuo@cgmh.org.tw (S.-F.K.); s843032@cgmh.org.tw (C.-Y.L.); 2Department of Medical Biotechnology and Laboratory Sciences, College of Medicine, Chang Gung University, Taoyuan 333, Taiwan; 3Division of Infectious Diseases, Department of Internal Medicine, Chiayi Chang Gung Memorial Hospital, Chiayi 613, Taiwan; r12045@cgmh.org.tw; 4School of Medicine, College of Medicine, Chang Gung University, Taoyuan 333, Taiwan; 5Microbiology Research and Treatment Center, Chiayi Chang Gung Memorial Hospital, Chiayi 613, Taiwan; 6Division of Infectious Diseases, Department of Internal Medicine, Kaohsiung Chang Gung Memorial Hospital, Kaohsiung 833, Taiwan

**Keywords:** rapid identification, bloodstream infections, sepsis, antimicrobial stewardship program, antibiotics

## Abstract

Timely pathogen identification in bloodstream infections is crucial for patient care. A comparison is made between positive blood culture (BC) pellets from serum separator tubes using a direct identification (DI) method and colonies on agar plates from a short-term incubation (STI) method with a matrix-assisted laser desorption/ionization Biotyper for the evaluation of 354 monomicrobial BCs. Both the DI and STI methods exhibited similar identification rates for different types of bacteria, except for Gram-positive and anaerobic bacteria. The DI method’s results aligned closely with the STI method’s results for Enterobacterales, glucose-non-fermenting Gram-negative bacilli (GNB), and carbapenem-resistant Enterobacterales. The DI method exhibited high concordance with the conventional method for GNB identification, achieving 88.2 and 87.5% accuracy at the genus and species levels, respectively. Compared with the STI method, the DI method showed a less successful performance for Gram-positive bacterial identification (50.5 vs. 71.3%; *p* < 0.01). The DI method was useful for anaerobic bacterial identification of slow-growing microorganisms without any need for colony growth, unlike in the STI method (46.7 vs. 13.3%; *p* = 0.04). However, both methods could not identify yeast in positive BCs. Overall, the DI method provided reliable results for GNB identification, offering many advantages over the STI method by significantly reducing the turnaround time and enabling quicker pathogen identification in positive BCs.

## 1. Introduction

Bloodstream infections, if neglected, may result in fatal consequences, underscoring the crucial need for timely identification of the responsible microorganisms and precise selection of suitable antimicrobial therapy to reduce both mortality and morbidity rates [1,2]. It is recommended to administer broad-spectrum antibiotics to patients suspected of harboring serious infections to minimize the risk of inadequate treatment. However, the extensive use of these broad-spectrum medications has been linked to the emergence of antimicrobial resistance and an increased incidence of adverse effects, such as allergic reactions, renal impairment, thrombocytopenia, *Clostridioides difficile* infection, and even higher mortality rates [2]. On the contrary, in certain cases, Gram-negative bacteria endowed with intrinsic resistance, like *Pseudomonas* sp., *Stenotrophomonas* sp., and *Acinetobacter* sp., have been spreading rapidly within the community, thereby heightening the potential for inadequate therapy [2]. This underscores the critical importance of utilizing antibiotics prudently to mitigate the risks associated with antimicrobial resistance and adverse events.

The conventional method utilized by bloodstream infections for detection is blood culture (BC), a pivotal task carried out in clinical laboratories, playing a crucial role in establishing an accurate etiological diagnosis [2]. Identifying the most common Gram-negative pathogens, such as *Escherichia coli*, *Klebsiella pneumoniae*, and *Pseudomonas aeruginosa*, is paramount in this process. Similarly, recognizing prevalent Gram-positive pathogens, like *Staphylococcus aureus*, *Enterococcus* spp., and *Streptococcus pneumoniae*, as well as yeasts including *Candida albicans*, *Candida parapsilosis*, *Candida krusei*, *Candida glabrata*, and *Cryptococcus neoformans*, is essential. The precise determination of the species of pathogen responsible for an infection is of utmost importance, given the necessity to promptly commence appropriate antimicrobial treatment. Moreover, the timeframe needed for diagnosis and identification holds significant importance, as any delay in this process can have a drastic impact on the prognosis of the patient [1,2]. It is crucial for clinical laboratories to efficiently and accurately identify pathogens causing bloodstream infections, as this information is vital for guiding effective treatment strategies and improving patient outcomes. Delays in the identification of pathogens can have serious consequences, such as inappropriate treatment, escalated healthcare expenses, and elevated mortality rates, emphasizing the crucial significance of prompt and precise detection techniques in clinical settings. A plethora of research endeavors have been dedicated to diminishing the time taken to identify pathogens in positive BCs. Typically, when BCs turn out positive, they are re-cultured on solid growth media. Following an incubation period of one night, the resulting colonies are analyzed utilizing matrix-assisted laser desorption/ionization time-of-flight mass spectrometry (MALDI-TOF MS) and subjected to antimicrobial susceptibility testing [1]. The utilization of MALDI-TOF MS within clinical laboratory settings has made a significant impact in recent years by notably reducing the duration needed to identify pathogens present in the bloodstream. Nevertheless, the process still takes approximately 12–24 h following the detection of BC positivity for the complete characterization of the pathogen, while even lengthier periods are essential before antimicrobial susceptibility testing results become accessible through conventional techniques. Despite the advancements facilitated by MALDI-TOF MS, the timeline for pathogen identification and subsequent antimicrobial susceptibility testing implementation in clinical laboratories remains a critical area that requires further improvement in order to enhance patient care and treatment outcomes. The primary drawback of utilizing this technique is the limitation in identifying the causative pathogen only after the growth and isolation of colonies, resulting in an extended turnaround time, particularly with slow-growing microorganisms, like anaerobic bacteria and yeast. It is crucial to promptly identify the causative microorganism for the proper administration of antimicrobial therapy to patients suffering from sepsis [2,3,4]. Additionally, an accurate and swift etiological diagnosis plays a pivotal role in reducing both the duration of hospitalization and the incurred costs [5,6,7]. Various efforts have been dedicated to the direct identification of microbes from positive BCs to diminish the time required for identification. Additionally, numerous subculture-independent methodologies have been devised for this purpose. Presently, there is a plethora of rapid molecular tests accessible for the direct identification (DI) of microbes from positive BC samples. However, the widespread application of these rapid molecular tests is hindered by the expensive nature of the testing process and the limited scope, as only a handful of species are covered in the test panels [8]. Another subculture-independent strategy revolves around the swift extraction of bacterial pellets from positive BCs. This particular method involves the elimination of non-microbial substances and microorganisms through centrifugation, thereby furnishing an adequate amount of biomass for the identification of pathogens utilizing MALDI-TOF MS. The acceleration of the incubation period for subculture agar plates significantly expedites the identification of pathogens [9,10,11].

Previous research has focused on the swift detection of bacteria straight from BC bottles using MALDI-TOF MS [3,4]. The time it takes to achieve final identification in this manner has been scrutinized in comparison to results obtained after leaving the samples overnight on solid media or subjecting them to short-term incubation (STI) on agar plates [1,12,13,14]. Various commercial kits, like Sepsityper1 (Bruker Daltonics, Bremen, Germany), Vitek MS BC (BioMerieux, Inc., Durham, NC, USA), and rapid BACpro1 II (Nittobo Medical Co., Tokyo, Japan) kits, have been utilized for rapid identification. However, these kits entail numerous complex procedures, often leading to suboptimal identification results and high costs [15]. Hence, in-house protocols have been implemented. Consequently, the present research aims to develop a straightforward, practical, and cost-efficient approach for promptly identifying pathogens in positive BCs. Furthermore, MALDI-TOF MS was employed to compare the rates of pathogen identification between the “in-house” DI and STI methods in positive BCs, while also exploring their potential implications for antibiotic treatment.

## 2. Materials and Methods

### 2.1. Microbiological Testing

The research was performed at the Chiayi Chung Gung Memorial Hospital in Central Taiwan. During January 2022 to December 2022, we collected 369 positive BCs from patients who were admitted to the hospital in a random manner; these were sent to the clinical microbiology laboratory. Each BC bottle that arrives to the clinical microbiology laboratory is incubated in the BACTEC™ FX system (BD Diagnostics, Sparks, MD, USA) for microorganism growth monitoring. Following microorganism growth identification by the BACTEC™ FX system, the blood culture media were collected from each bottle and subjected to Gram staining. These BCs were selected only based on Gram staining in order to confirm the presence of one organism per BC bottle for further study. All of these BCs were collected in Plus Aerobic and Anaerobic/F Culture Vials (BD Diagnostics, Sparks, MD, USA). Only one sample per patient was included in the study to avoid repetition. This process ensured the accuracy and reliability of the microbial data collected for the study, as repetitive samples from the same patient could skew the results and lead to erroneous conclusions.

For conventional testing, a small amount of the positive specimen was placed on various agar media, such as blood/EMB, chocolate, and CDC ANA, followed by an incubation period at a temperature of 37 °C s in an environment with 5% carbon dioxide concentration for a duration of 18 to 24 h. Additionally, CDC ANA media were incubated at 37 °C for 5 days, under anaerobic conditions. Subsequently, the process of identifying the microorganisms at the species level was carried out utilizing the MALDI-TOF MS Biotyper system manufactured by Bruker Daltonics GmbH, known as the Bruker Microflex LT/SH. In addition to conventional method, the DI and STI methods were performed simultaneously after obtaining the Gram staining results.

### 2.2. DI Method

DI was performed using an in-house pretreatment procedure. Blood was drawn from BCs into 5 mL BD Vacutainer Blood Collection separator tubes. The tubes contain spray-coated silica and a polymer gel. Its main function is to separate the serum or plasma from the blood cells during centrifugation, maintaining the purity and stability of the sample, thus facilitating accurate clinical tests and analyses. Consequently, these tubes were spun at room temperature at 1720× *g* for a period of 10 min. Upon the removal of the supernatant, 1 mL of distilled water was methodically and slowly introduced to delicately resuspend the pellet without causing any disruption to the gel layer. The resulting mixture was then carefully transferred to a 1.5 mL reaction tube (Eppendorf, Hamburg, Germany) and subjected to further centrifugation at room temperature at a speed of 9178× *g* for 5 min. A dropper was skillfully employed to eliminate a significant portion of the supernatant, followed by the use of a pipette to meticulously cleanse the remaining sample. The sediment obtained from this process was meticulously applied to the MALDI-TOF sample target for subsequent analysis. Each individual sample underwent testing in duplicate to ensure reliability, with only the spot exhibiting the highest probability score for identification being taken into consideration, as depicted in Figure 1. This thorough and systematic pretreatment procedure played a crucial role in guaranteeing that the samples were suitably primed for precise identification, thereby significantly reducing the likelihood of potential contamination or inadvertent loss of microbial material.

### 2.3. STI Method

The process of species identification was carried out subsequent to a brief incubation period [10,11], during which time bacterial colonies on blood agar were allowed to proliferate. Subsequently, after 6 h of incubation, a thin layer of these colonies was transferred onto a target plate (Bruker Daltonics in Bremen, Germany) for MALDI-TOF MS analysis to determine the species of the bacterial isolates (Figure 1).

### 2.4. MALDI-TOF MS Analysis

Following drying of spotted colonies (conventional testing and the STI method) or the bacterial pellet (the DI method) on a MALDI-TOF MS target plate (Bruker Daltonics, Bremen, Germany), 1 μL of 70% formic acid was placed on the sample target and allowed to dry at room temperature before 1 μL of alpha-cyano-4-hydroxycinnamic acid (HCCA) matrix solution was placed onto each spot and air-dried. MALDI-TOF MS was conducted using the microflex LT/SH MALDI-TOF mass spectrometer (Bruker Daltonics GmbH & Co. KG, Bremen, Germany) with MBT Compass V4.1, and the measured protein profiles were compared with the MBT Compass Library and Revision K MBT 7311 MSP Library databases. The calibration and validation of MALDI-TOF MS were carried out daily using a bacterial test standard according to the manufacturer’s instructions, and using *S. aureus* ATCC25923, *B. fgagilis* ATCC25285, and *Candida tropicalis* for quality control. MALDI-TOF MS results were interpreted according to the manufacturer’s technical specifications as follows: a score <1.7 indicates no reliable identification, a score between 1.7 and 1.999 indicates identification to the genus level, and a score ≥2 indicates identification to the species level.

Positive BC bottles were subjected to pretreatment, which was mentioned before to obtain the bacterial pellets. Then, the DI method with MALDI-TOF was used to identify the bacterial strains. After a 6 h culture, bacterial colonies were subjected to MALDI-TOF for bacterial strain identification using the STI method. In the conventional method, the culture medium was incubated for an additional 12–18 h, followed by MALDI-TOF for bacterial strain identification. The results of the conventional method were used as the final identification results to compare the identification efficiencies of the DI and STI methods.

### 2.5. Statistical Analyses

Statistical analyses were performed using the Chi-square and Fisher’s exact tests, selecting the appropriate method based on the characteristics of the data. Statistical significance was set at *p* < 0.05. Identification results of the conventional method were obtained and compared with those of the DI and STI methods to evaluate and validate their efficiency and accuracy.

## 3. Results

Here, 369 BC bottles were evaluated for microbial growth using the BD BACTEC FX instrument (BD Diagnostics, Sparks, MD, USA). After excluding 15 BC bottles with polymicrobial growth, 354 bottles with monomicrobial growth were included in this study. The collection of samples consisted of 167 isolates (47.2%) of Gram-positive bacteria, 152 isolates (42.9%) of Gram-negative bacteria, 19 isolates (5.4%) of yeast, and 16 isolates (4.5%) of anaerobic bacteria. The process of identification using the DI method takes approximately 1 h, while the STI method requires around 6 h for completion. In contrast, the conventional method requires 18 to 24 h for the identification of microbes in positive BCs. The mean time interval between bacterial growth detection as reported by the BC incubation system and the results obtained through the STI method was 6 ± 1.5 h in general. Conversely, the average duration associated with the DI method was 1 ± 0.4 h. The period required for pathogen identification in positive BCs was notably reduced by 5 h when employing our in-house DI method as opposed to the STI method.

Out of the 354 samples analyzed, 264 (74.6%) and 224 (63.3%) were identified with a score equal to or greater than 1.7 using the STI and DI methods, respectively. The results obtained from the DI method for positive BCs are graphically represented in Figure 2. Among the 231 strains (65.3%) that were reliably identified, 225 (63.6%) were accurately classified at the species level, while 4 (1.1%) were correctly categorized at the genus level, and 2 (0.6%) were misclassified. Notably, the absence of a peak in the MALDI-TOF MS analysis of 123 samples utilizing the DI method resulted in a failure to identify 78 Gram-positive bacteria (63.4%), 18 Gram-negative bacteria (14.6%), 8 anaerobic bacteria (6.5%), and 19 yeasts (15.4%).

When applying the DI method (Table 1), 134 Gram-negative bacteria (88.2%), 8 anaerobes (50.0%), and 83 Gram-positive bacteria (49.7%) were successfully identified. Interestingly, none of the yeast isolates present in the positive BC samples were identified through the use of the DI method. Table 2 provides a detailed account of the identification outcomes for Gram-negative bacteria in the positive BCs analyzed via MALDI-TOF MS with the DI method, along with their corresponding analysis scores. The predominant strains identified included *Escherichia coli* (*n* = 85), *Klebsiella pneumoniae* (*n* = 18), and *Pseudomonas aeruginosa* (*n* = 8), exhibiting identification rates of 90.6%, 88.9%, and 87.5%, respectively. Furthermore, Table 3 offers insights into the identification results of Gram-positive bacteria in the positive blood cultures examined through MALDI-TOF MS using the DI method, alongside their analysis scores. The most prevalent strains detected were *Staphylococcus aureus* (*n* = 31), *S. epidermidis* (*n* = 28), and *S. capitis* (*n* = 22), with corresponding identification rates of 51.6%, 50.0%, and 68.2%, respectively.

All 134 Gram-negative bacteria displaying peaks in the MALDI-TOF MS analysis utilizing the DI method were accurately identified at the species level, as detailed in Table 2. Among the subset of 89 Gram-positive bacteria exhibiting peaks in the MALDI-TOF MS analysis with the DI method, 83 (93.3%) were precisely identified at the species level. At the genus level, only four Gram-positive bacteria were correctly identified, while two strains (*S. capitis* and *Streptococcus oralis*) were erroneously classified, as indicated in Table 3.

Figure 3 presents a comparison of the rates at which DI and STI methods identify different bacterial species. It is worth noting that there were no statistically significant variances in the identification rates between the two methods when it came to Gram-negative bacteria, such as Enterobacterales, glucose-non-fermenting Gram-negative bacilli, and carbapenem-resistant Enterobacterales. On the other hand, for Gram-positive bacteria, particularly *S. aureus* and coagulase-negative staphylococci, the identification rate of the STI method (71.3% and 75.6%, respectively) was significantly higher compared to the DI method (50.5% and 59.3%, respectively) with a *p*-value of less than 0.01 for both cases. In contrast, the DI method outperformed the STI method in identifying anaerobic bacteria, with rates of 46.7% and 13.3%, respectively, and a *p*-value of 0.04. However, it is important to note that both methods demonstrated suboptimal performance in yeast identification. The DI method failed to identify any yeast isolates, while the STI method only managed to identify 10.5% of the isolates.

## 4. Discussion

Identifying the causative agent is essential in cases of bloodstream infection due to the fact that any delay in the initiation of antimicrobial therapy has been linked to heightened unfavorable outcomes for patients and elevated mortality rates, especially in instances where microorganisms with inherent resistance, such as *Pseudomonas* sp. and *Stenotrophomonas* sp., are identified. The precise detection of the pathogens plays a critical role as inaccurate identification outcomes and inadequate microbial information can lead to the inappropriate utilization of antibiotics, thereby triggering the emergence of antimicrobial resistance and/or disruption of the normal balance of microorganisms within the microbiota [2]. It is imperative to swiftly and accurately determine the etiologic pathogen in bloodstream infection cases to ensure timely administration of the appropriate antimicrobial agents, which can significantly impact patient prognosis and decrease mortality rates [1]. Failure to promptly identify the causative agent may result in prolonged illness, increased healthcare costs, and a greater risk of treatment failure, underscoring the importance of rapid and accurate diagnostic approaches in the management of bloodstream infections. Furthermore, the identification of pathogens can guide clinicians in selecting the most effective treatment strategies tailored to the specific infectious agent, thereby optimizing patient care and enhancing clinical outcomes. Inaccurate or delayed identification of pathogens can lead to suboptimal treatment regimens, prolonged hospital stays, and increased susceptibility to complications, emphasizing the critical role of accurate pathogen identification in improving patient outcomes and reducing the burden of bloodstream infections. Additionally, the emergence of multidrug-resistant organisms underscores the importance of accurate pathogen identification in guiding appropriate antibiotic therapy and implementing infection control measures to prevent the spread of resistant strains in healthcare settings.

In this study, we assessed the efficacy of pathogen detection using DI or STI techniques through MALDI-TOF MS. Our findings indicated that both the DI and STI approaches demonstrated comparable levels of identification accuracy across a range of bacterial strains, except for Gram-positive bacteria, such as *S. aureus* and coagulase-negative staphylococci. In particular, 87 (52.1%) isolates out of 167 samples of Gram-positive bacteria could be recognized utilizing the DI method. On the other hand, 134 (88.2%) isolates out of 152 samples of Gram-negative bacteria were identified when applying the DI method (Table 1). Adequate cell disruption plays a crucial role in the efficient extraction of intracellular proteins and the production of high-quality mass spectra. The cell disruption methods employed in our in-house DI approach may prove inadequate for Gram-positive bacteria, resulting in suboptimal protein extraction efficiency and, consequently, lower identification performance. Accurate identification of Gram-negative bacteria is necessary to identify the most suitable antibiotics, whereas phenotypic tests can be instrumental in identifying the causative agent in cases of Gram-positive bacteremia, thereby facilitating more targeted antibiotic therapy with a reduced risk of resistance development. In contrast, the DI method demonstrated superior performance for anaerobic bacteria from positive BCs compared to the STI method. Owing to the slow growth of anaerobic bacteria, they do not produce colonies, making identification less effective with the STI method (Figure 3). However, the DI method allows detection without waiting for growth, shortening the turnaround time for anaerobic bacteria identification by at least 48–72 h [16]. Therefore, in patients with sepsis where anaerobic bacterial bloodstream infections are suspected, rapid diagnosis by the DI method followed by prompt treatment are crucial to avoid serious complications [17,18]. The DI method for Gram-negative bacterial identification from positive BCs showed good and reliable results, with concordance rates of 88.2% at the genus level and 87.5% at the species level, compared to the conventional method (incubation for 18–24 h). Comparing the pathogen identification performance between the DI and STI methods, there were no significant differences in the identification of Gram-negative bacteria, such as Enterobacterales, glucose-nonfermenting Gram-negative bacilli, including *Pseudomonas* sp. and *Stenotrophomonas* sp., or even carbapenem-resistant Enterobacterales strains from positive BCs (all *p* > 0.5). In contrast, the STI method demonstrated superior performance for Gram-positive bacterial identification from positive BCs compared to the DI method. Furthermore, these two methods did not identify yeast-positive BCs. Among Gram-negative bacteria, Enterobacterales had the highest identification rate, whereas the DI method yielded unsatisfactory results for the identification of Gram-positive bacteria and yeasts in positive BCs, with no identification at the species level, which are difficult to identify via MS analysis [19]. Successful microbial identification in the DI assay depends on the number of bacteria collected from the culture pellet. The thick cell walls of Gram-positive bacteria and yeast make it challenging to identify the culprit microorganisms using rapid pellets prepared from positive BCs [4,19].

Accurate identification is validated by achieving a score ≥1.7 in MALDI-TOF MS. The DI method employed in this study exhibited a high pathogen identification rate of 88.2% for Gram-negative bacteria, 50% for anaerobic bacteria, and 49.7% for Gram-positive bacteria, showcasing its effectiveness across different bacterial types. Just like various other laboratory-developed methodologies, our “in-house” DI method stands out for its efficiency, speed, and simplicity in execution, as highlighted in previous studies [20,21]. In contrast to methods that require intricate buffer preparation and numerous centrifugation steps, our method simplifies the process by only needing two centrifugation steps and the addition of distilled water, making it more user-friendly and less time-consuming for laboratory technicians. A comparative analysis with commercially available kits reveals that our “in-house” DI method not only provides accurate results but also delivers significant time- and cost savings, a crucial factor for many healthcare settings [6,20,21]. Swift and precise identification of pathogens found in BCs is particularly crucial for individuals experiencing sepsis, a life-threatening condition, as it directly impacts morbidity and mortality rates linked to sepsis [22]. Traditionally, the identification of blood pathogens involves cultivation and subsequent identification through conventional methods, a process that typically takes around 18 to 24 h to yield results. This prolonged duration often necessitates the initiation of empirical antibiotic therapy or withholding treatment altogether during this diagnostic window, underscoring the urgent need for faster diagnostic approaches. Decreasing the duration required for pathogen identification offers the prospect of enhancing patient outcomes through enabling targeted antimicrobial therapy and potentially reducing the unnecessary use of broad-spectrum antibiotics.

This research has certain limitations that need to be addressed. The sample sizes for the sub-groups of yeast and anaerobic bacteria were relatively small, and the analysis was limited to monomicrobial blood cultures. Therefore, it is imperative to conduct additional studies to validate whether the DI method outperforms the STI method in detecting anaerobic bacteria. It is worth noting that the manufacturer recommended the addition of 1 μL of formic acid after the deposition of microbial material on the MS target for the identification of Gram-positive bacteria and yeasts, which could have potentially impacted the quality of identification in this particular study. The utilization of formic acid on the MS target was executed through both DI and STI methodologies within the scope of this investigation. An alternative strategy, such as physical disruption (e.g., sonication) or chemical disruption (e.g., enzyme treatment), to improve protein extraction should address these issues. Although MALDI-TOF MS successfully identified the pathogens in the majority of cases, it is crucial to ensure the purity of blood cultures, regardless of the identification method employed. While the DI method proved to be valuable for pathogen identification, it did not show efficacy in antimicrobial susceptibility testing. The knowledge of local antibiograms of microorganisms, especially those with intrinsic resistance, such as *Pseudomonas* sp. and *Stenotrophomonas* sp., can significantly aid in treatment management. The swift identification of pathogens can assist clinicians in choosing the most appropriate antimicrobial therapy [23,24]; nonetheless, further investigations are necessary to explore the impact of the DI method on the clinical outcomes of patients. Despite molecular assays being the most efficient for pathogen detection in BCs, their limited usage is primarily attributed to their high cost. Additionally, these tests’ panels can only detect specific species [25,26,27,28,29,30]. Hence, our DI method proves to be more efficient than these molecular assays.

## 5. Conclusions

Our in-house DI method offers a rapid and easy-to-use alternative for identifying pathogens in BCs within just one hour, leading to a substantial time-saving of five hours compared to the STI technique commonly used in clinical microbiology laboratories. The efficacy of the DI method closely corresponds to that of the STI method, particularly in the case of Gram-negative bacteria. However, the cell disruption techniques utilized in our in-house DI strategy may not be sufficient for Gram-positive bacteria and yeasts, resulting in suboptimal extraction of proteins and, subsequently, lower identification accuracy. Nevertheless, the DI method involves utilizing BC pellets in conjunction with MALDI-TOF and Gram-staining for the swift and precise diagnosis of bloodstream infections. By eliminating the need for incubation, this method facilitates the prompt identification of bacteria in samples without incurring additional costs, which is particularly crucial for urgent diagnostic procedures.

## Figures and Tables

**Figure 1 diagnostics-14-01611-f001:**
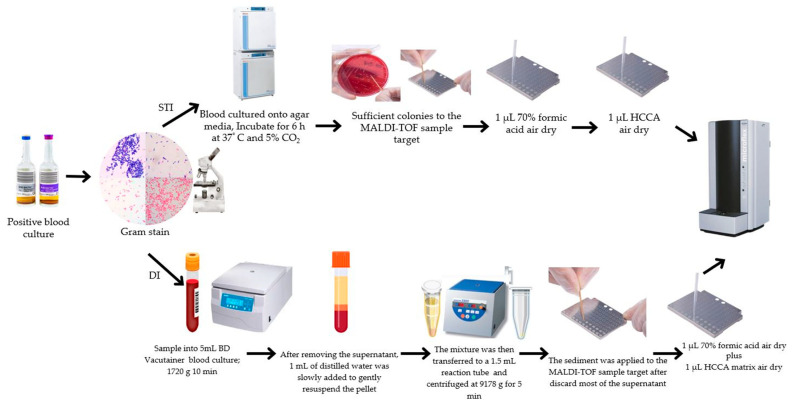
Flow chart of the short-term incubation (STI) method and the direct identification (DI) method.

**Figure 2 diagnostics-14-01611-f002:**
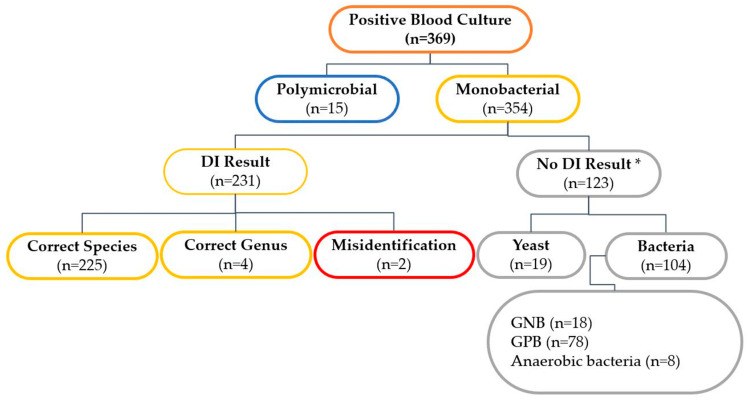
Direct identification (DI) method results from positive blood cultures. GNB, Gram-negative bacteria; GPB, Gram-positive bacteria. * No peak in MALDI-TOF MS analysis of the DI method.

**Figure 3 diagnostics-14-01611-f003:**
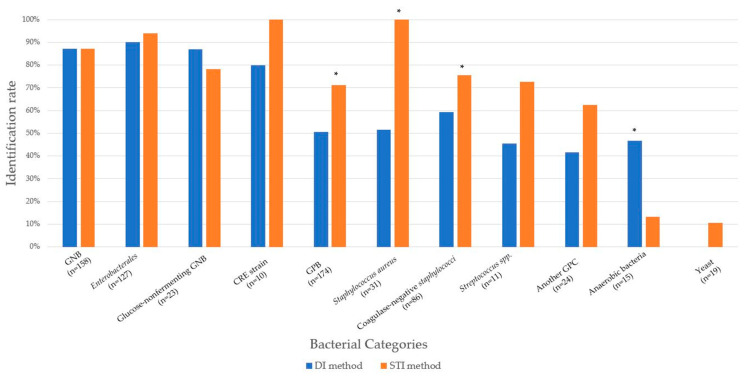
Comparison of identification rate between the direct identification (DI) and short-term incubation (STI) methods. CRE, carbapenem resistant Enterobacterales; GNB, Gram-negative bacteria; GPB, Gram-positive bacteria; GPC, Gram-positive cocci; *n*, tested number. * *p* < 0.05 indicates significant difference.

**Table 1 diagnostics-14-01611-t001:** Results of MALDI-TOF MS analysis for microorganism identification with direct identification method are divided according to microorganism groups.

Microorganism	N	Direct Identification Method					
		Score ^a^ < 1.7	1.7 ≤ Score ^a^ ≤ 1.99	Score ^a^ ≥ 2.0	Correct Genus	Misdentification	No Identification
Gram-negative bacteria	152	1	5	128	0	0	18
Gram-positive bacteria	167	0	46	37	4	2	78
Yeast	19	0	0	0	0	0	19
Anaerobic bacteria	16	0	2	6	0	0	8
Total N (%)	354	1 (0.3)	53 (15.0)	171 (48.3)	4 (1.1)	2 (0.6)	123 (34.7)

N, number. ^a^ MALDI-TOF MS score.

**Table 2 diagnostics-14-01611-t002:** Number of isolates from the direct identification results of Gram-negative bacteria.

Species	Correct at Species Level			Correct at Genus Level			Misidentified			Non-Reliable Identification (No DI Result *)	Total Correct Identification (Identification Rate %)
	MALDI-TOF MS Score			MALDI-TOF MS Score			MALDI-TOF MS Score				
	<1.7	1.7–1.99	≥2	<1.7	1.7–1.99	≥2	<1.7	1.7–1.99	≥2		
*Acinetobacter baumannii*			1								1 (100)
*Acinetobacter johnsonii*			1								1 (100)
*Acinetobacter ursingii*			1								1 (100)
*Achromobacter xylosoxidans*										2	0
*Alistipes onderdonkii*			1								1 (100)
*Aeromonas caviae*		1									1 (100)
*Aeromonas veronii*			1								1 (100)
*Bacteroides fragilis*			2								2 (100)
*Bacteroides thetaiotaomicron*			1								1 (100)
*Burkholderia multivorans*		1									1 (100)
*Citrobacter diversus*			3								3 (100)
*Citrobacter freundii complex*			1								1 (100)
*E. coli*			77							8	77 (90.6)
*Elizabethkingia meningoseptica*			1								1 (100)
*Enterobacter cloacae complex*			5							1	5 (83.3)
*Herbaspirillum aquaticum*			1								1 (100)
*Herbaspirillum huttiense*	1										1 (100)
*Klebsiella pneumoniae*			16							2	16 (88.9)
*Moraxella osloensis*										1	0
*Moraxella* sp.										1	0
*Morganella morganii*			1							1	1 (50.0)
*Pantoea dispersa*			1								1 (100)
*Parabacteroides distasonis*										1	0
*Prevotella* sp.										1	0
*Proteus mirabilis*			1								1 (100)
*Providencia stuartii*				1						1	1 (50.0)
*Pseudomonas aeruginosa*			7							1	7 (87.5)
*Ralstonia mannitolilytica*		1									1 (100)
*Salmonella* sp.			5								5 (100)
*Serratia marcescens*			2								2 (100)
*Sphingomonas parapaucimobilis*		1									1 (100)
*Stenotrophomonas maltophilia*		1	3								4 (100)

* No peak in MALDI-TOF MS analysis of the direct identification method.

**Table 3 diagnostics-14-01611-t003:** Number of isolates from the direct identification results of Gram-positive bacteria.

Species	Correct at Species Level			Correct at Genus Level			Misidentified			Non-Reliable Identification (No DI Result *)	Total Correct Identification (Identification Rate %)
	MALDI-TOF MS Score			MALDI-TOF MS Score			MALDI-TOF MS Score				
	<1.7	1.7–1.99	≥2	<1.7	1.7–1.99	≥2	<1.7	1.7–1.99	≥2		
*Aerococcus*										1	0
*Arthrobacter creatinolyticus*			1								1 (100)
*B-Streptococcus Group B*		1	2								3 (100)
*Bacillus*		1								2	1 (33.3)
*Bacillus flexus*										1	0
*Bacillus subtilis*					1						1 (100)
*Bacillus horneckiae*			1								1 (100)
*Cellulosimicrobium cellulans*			1								1 (100)
*Clostridium clostridioforme*										1	0
*Coagulase (−) Staphylococcus*										1	0
*Corynebacterium imitans*										1	0
*Corynebacterium propinquum*										2	0
*Corynebacterium* sp.										3	0
*Enterococcus faecalis*		2	2								4 (100)
*Enterococcus faecium*		1	3			1				3	5 (62.5)
*Enterococcus gallinarum*										1	0
*Enterococcus raffinosus*										1	0
*Lactobacillus rhamnosus*										1	0
*Lactococcus garvieae*										1	0
*Leuconostoc*										1	0
*Microbacterium* sp.										2	0
*Micrococcu*										2	0
*Micrococcus luteus*										2	0
*Paenibacillus* sp.										1	0
*Propionibacterium acnes*			1							2	1 (33.3)
*Propionibacterium* sp.		2								2	2 (50.0)
*Staphylococcus aureus*		8	8							15	16 (51.6)
*Staphylococcus capitis*		8	6			1			1	6	15 (68.2)
*Staphylococcus caprae*		2									2 (100)
*Staphylococcus epidermidis*		12	2							14	14 (50.0)
*Staphylococcus haemolyticus*		2	4		1					5	7 (58.3)
*Staphylococcus hominis*		5	4							5	9 (64.3)
*Staphylococcus pettenkoferi*		2									2 (100)
*Staphylococcus saprophyticus*										2	0
*Staphylococcus warneri*			3							1	3 (75.0)
*Streptococcus constellatus*										1	0
*Streptococcus cristatus*										1	0
*Streptococcus mitis*		1								1	1 (50.0)
*Streptococcus oralis*		1						1			1 (50.0)
*Streptococcus salivarius*										1	0
*Staphylococcus sciuri*			1								1 (100)
*Viridans streptococcus*										1	0

* No peak in MALDI-TOF MS analysis of the direct identification method.

## Data Availability

Derived data supporting the findings of this study are available from the first author Shu-Fang Kuo upon request.

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
