# Peer review of "Comparison of the Direct Identification and Short-Term Incubation Methods for Positive Blood Cultures via MALDI-TOF Mass Spectrometry"

_diagnostics, 2024, doi:10.3390/diagnostics14151611_

Round 1

Reviewer 1 Report

Comments and Suggestions for Authors

Abstract

The authors should add a context/state of the art before the objectives. Results should be shortened. The need to develop new method should be added, highlighted.

Introduction

The authors should highlight the need of new method and the fact that they have developed the “in-house” DI.

Materials and methods

Why the authors performed only one sample per patient?

Why they incubated all specimen at 37°C and in a 5% CO2 environment since bacteria do not have the same optimal growth temperature ?

DI method: the authors must give the centrifuge speed in g and not in rpm. They must indicate the temperature at which they centrifuge.

What is the gel layer? to which biological elements it correspond ?

The figure 1 is not in adequation with the informations within the text (temperature of incubation, lack of informations about the results of the second centrifuge in di method).

In case of already published methods for STI &/or DI, references should be added with informations about optimizations or changes pointed out.  

High speed centrifugation: at which speed? for how long? at which temperature??

Results

How the authors explain the absence of a peak while using the DI method which seems to represent one drawback of their method (at least 1/3 of results)

The authors should explain the difference scores and their relevance in the method used.

The tables 2 & 3 must be redrawn to fit on only one page each, in a portrait mode. They should contain the same column in order to compare the results obtained.

The table 4 should be replaced by a histogram to allow better comparisons between both DI & STI methods.

Discussion

L32-40 : the lines should be added in the abstract part

L52: DI was able to detect gram+ bacteria (table 1). Could you detail this point ?

L78-l80: how could the authors overcome the problem encountered with gram-positive bacteria & yeasts?

L103-109 : the sentences should be placed as a conclusion part

L120-121 : why the authors and how they evaluated DI method for anti-microbial susceptibility?

Conclusion

L132 : does the duration of 5-hour valuable in every case or only for shorter ones ? this point should be quantified in result part and discussed afterwards.

Author Response

We have provided our point-by-point responses to the comments below. The changes in the revised edition are highlighted in red color. We hope you will find our revised manuscript suitable for publication in Diagnostics.

Reviewers' comments:

Reviewer #1:

Abstract

The authors should add a context/state of the art before the objectives. Results should be shortened. The need to develop new method should be added, highlighted.

Response: We appreciate the constructive comment from the reviewer. We rewrite the Abstract as “Timely pathogen identification in bloodstream infections is crucial for patient care. Comparison made between positive blood culture (BC) pellets from serum separator tubes using direct identification (DI) method and colonies on agar plates from short-term incubation (STI) method with matrix-assisted laser desorption/ionization Biotyper for the evaluation of 354 monomicrobial BCs.” (page 1, lines 16-20)

Introduction

The authors should highlight the need of new method and the fact that they have developed the “in-house” DI.

Response: We appreciate the suggestion from the reviewer. The need of new method and the fact that they have developed the “in-house” DI was put in the introduction part. (page 2-3, line 87-114)

Materials and methods

Why the authors performed only one sample per patient?

Response: A patient may acquire a recurrent bacterial infection within a brief period. Consequently, a singular sample is obtained from each patient. As a result, only one sample per patient was included in the study to avoid repetition. (page 3, line 127)

Why they incubated all specimen at 37°C and in a 5% CO2 environment since bacteria do not have the same optimal growth temperature?

Response: The primary clinical goal is to rapidly and accurately identify pathogens. The conditions of 37°C and 5% CO2 generally achieve this for most cases. While not optimal for all bacteria, these conditions cover the growth needs of most clinically relevant pathogens and provide a standardized, practical, and efficient operational environment. To be clarified, we add a sentence “Additionally, CDC ANA media were incubated at 37°C for 5 days, under anaerobic conditions.” (page 3, line 134-135)

DI method: the authors must give the centrifuge speed in g and not in rpm. They must indicate the temperature at which they centrifuge.

Response: We appreciate the constructive comment from the reviewer. We rewrite the sentence as “these tubes were spun at room temperature at 1,720 g for a period of 10 mins.” (page 3, line 143) and “subjected to further centrifugation at room temperature at a speed of 9,178 g for 5 mins.” (page 3, line 147-148)

What is the gel layer? to which biological elements it corresponds?

Response: We used BD Vacutainer Blood Collection separator tubes in the DI method. BD Vacutainer® SST™ Tubes contain spray-coated silica and a polymer gel. Its main function is to separate the serum or plasma from the blood cells during centrifugation, maintaining the purity and stability of the sample, thus facilitating accurate clinical tests and analyses.

The figure 1 is not in adequation with the information within the text (temperature of incubation, lack of information about the results of the second centrifuge in di method).

Response: We appreciate the suggestion from the reviewer. It is recommended to retain the existing figure 1 for a succinct depiction. Elaborate information has been included in the textual content. However, the centrifuge speed is presented in g instead of in rpm.

In case of already published methods for STI &/or DI, references should be added with information about optimizations or changes pointed out.  

Response: We add the reference [10,11] for STI method. (page 4, line 162) DI was performed using an in-house pretreatment procedure.

High speed centrifugation: at which speed? for how long? at which temperature??

Response: The centrifuge speed, timing and temperature were prescribed at DI method. (page 3, line 142-148) To be clarified, we rewrite the sentence as “Positive BC bottles were subjected to pretreatment, including cell removal and high-speed centrifugation, which was mentioned before to obtain the bacterial pellets.” (page 5, line 181-182)

Results

How the authors explain the absence of a peak while using the DI method which seems to represent one drawback of their method (at least 1/3 of results)

Response: We appreciate the constructive comment from the reviewer. The absence of peaks when using the direct blood bottle identification method almost always occurs with Gram-positive bacteria and yeasts. The possible reason is the cell wall structure. Gram-positive bacteria have a thicker cell wall mainly composed of peptidoglycan, which makes protein extraction and analysis difficult. In MALDI-TOF analysis, the mass spectra of proteins are crucial for identifying bacteria; thus, the thick cell wall affects the quality and clarity of the mass spectra. Effective cell disruption is key to successfully extracting internal proteins and generating high-quality mass spectra. Existing cell disruption techniques, such as physical disruption (e.g., sonication) or chemical disruption (e.g., enzyme treatment), may be insufficient for Gram-positive bacteria and yeasts, leading to low protein extraction efficiency. Improving protein extraction techniques should address these issues. To highlight the issue, we rewrite the phrase It is worth noting that the manufacturer recommended the addition of 1 μL of formic acid post the deposition of microbial material on the MS target for the identification of Gram-positive bacteria and yeasts, which could have potentially impacted the quality of identification in this particular study. The utilization of formic acid on the MS target was executed through both DI and STI methodologies within the scope of this investigation. An alternative strategy improving protein extraction techniques should address these issues.” (page 17, line 111-117)

The authors should explain the difference scores and their relevance in the method used.

Response: MALDI-TOF MS results were interpreted according to the manufacturer's technical specifications as follows: score < 1.7 indicates no reliable identification, a score between 1.7 and 1.999 indicates identification to the genus level, and a score ≥ 2 indicates identification to the species level. (page 4-5, line 177-180)

The tables 2 & 3 must be redrawn to fit on only one page each, in a portrait mode. They should contain the same column in order to compare the results obtained.

Response: We appreciate the suggestion from the reviewer. We revised the Table 2 & 3 contain the same column to compare the results obtained.

The table 4 should be replaced by a histogram to allow better comparisons between both DI & STI methods.

Response: Thank you for your valuable feedback. We propose retaining the present format of Table 4 owing to the presence of 11 strains and the diverse samples acquired for each strain.

Discussion

L32-40: the lines should be added in the abstract part

Response: We appreciate the suggestion from the reviewer. We rewrite the Abstract as “Timely pathogen identification in bloodstream infections is crucial for patient care. Comparison made between positive blood culture (BC) pellets from serum separator tubes using direct identification (DI) method and colonies on agar plates from short-term incubation (STI) method with matrix-assisted laser desorption/ionization Biotyper for the evaluation of 354 monomicrobial BCs.” (page 1, lines 16-20)

L52: DI was able to detect gram+ bacteria (table 1). Could you detail this point?

Response: We appreciate the constructive comment from the reviewer. To be clarified, we rewrite the phrase as “Our findings indicated that both DI and STI approaches demonstrated comparable levels of identification accuracy across a range of bacterial strains, except for Gram-positive bacteria, such as S. aureus and coagulase-negative staphylococci. In particular, 78 (46.7%) isolates out of 167 samples of Gram-positive bacteria could not be recognized utilizing the DI method. On the other hand, 18 (11.8%) isolates out of 152 samples of Gram-negative bacteria remained unidentified when applying the DI technique (Table 1).” (page 16, line 51-57)

L78-l80: how could the authors overcome the problem encountered with gram-positive bacteria & yeasts?

Response: We appreciate the suggestion from the reviewer. Effective cell disruption is key to successfully extracting internal proteins and generating high-quality mass spectra. Existing cell disruption techniques, such as physical disruption (e.g., sonication) or chemical disruption (e.g., enzyme treatment), may be insufficient for Gram-positive bacteria and yeasts, leading to low protein extraction efficiency. Improving protein extraction techniques should address these issues. We rewrite the phrase asIt is worth noting that the manufacturer recommended the addition of 1 μL of formic acid post the deposition of microbial material on the MS target for the identification of Gram-positive bacteria and yeasts, which could have potentially impacted the quality of identification in this particular study. The utilization of formic acid on the MS target was executed through both DI and STI methodologies within the scope of this investigation. An alternative strategy improving protein extraction techniques should address these issues.” (page 17, line 111-117)

L103-109: the sentences should be placed as a conclusion part

Response: Thank you for your valuable feedback. The sentence has been placed as a conclusion part. (page 18, line 127-132)

L120-121: why the authors and how they evaluated DI method for anti-microbial susceptibility?

Response: We appreciate the constructive comment from the reviewer. The phrase “While the DI method proved to be valuable for pathogen identification, it did not show efficacy in antimicrobial susceptibility testing. The knowledge of local antibiograms of microorganisms, especially those with intrinsic resistance such as Pseudomonas sp. and Stenotrophomonas sp., can significantly aid in treatment management. The swift identification of pathogens can assist clinicians in choosing the most appropriate antimicrobial therapy [23,24]; nonetheless, further investigations are necessary to explore the impact of the DI method on the clinical outcomes of patients.” was put in the Discussion part. (page 17, line 119-125)

Conclusion

L132: does the duration of 5-hour valuable in every case or only for shorter ones? this point should be quantified in result part and discussed afterwards.

Response: We appreciate the precise comments from the reviewer. To be clarified, we rewrite the sentence “The mean time interval between bacterial growth detection as reported by the BC incubation system and the results obtained through the STI method was 6±1.5 hours in general. Conversely, the average duration associated with the DI method was 1±0.4 hours.” in result part. (page5, line 203-206) The key benefit of employing the DI technique is its ability to significantly diminish the time required for bacterial identification by 5 hours in comparison to the STI method. This time-saving aspect is particularly critical in scenarios necessitating prompt identification and commencement of suitable antibiotic treatment to influence patient outcomes, especially in instances of severe sepsis or other life-threatening infections.

Reviewer 2 Report

Comments and Suggestions for Authors

It is an interesting study that considers certain aspects of public health, namely the presence of bacteria in clinical samples. Specific methods exist to establish the etiological diagnosis of blood infections. The present work refers to a modern method. However, from my point of view, several minor corrections are needed in the manuscript to enhance its potential for publication.

Lines 36-39: more bibliographic references are needed

Why did you choose to use Clostridioses difficilae instead of Clostridium difficilae?

Lines 40-44: It is a bibliographic reference study

Lines 44, 52, 54: Please correct the names of gram-negative and gram-positive bacteria. The entire manuscript needs correction.

Lines 61-63: Please insert bibliographical references

Lines 101-102: Who conducted this study?

No information is mentioned about the spread of this method in clinical diagnostic laboratories, its costs, or its profitability.

From my point of view, the number of samples processed in a year is relatively low, especially since it is a hospital. What is the capacity of the hospital under study?

Author Response

We have provided our point-by-point responses to the comments below. The changes in the revised edition are highlighted in red color. We hope you will find our revised manuscript suitable for publication in Diagnostics.

Reviewers' comments:

Reviewer #2:

Lines 36-39: more bibliographic references are needed

Response: Thank you for your suggestion. New bibliographic references have been added. (line 39)

Why did you choose to use Clostridioses difficilae instead of Clostridium difficilae?

Response: The species was transferred from the genus Clostridium to Clostridioides in 2016, thus giving it the binomial Clostridioides difficile. (Zhu D, Sorg JA, Sun X. Clostridioides difficile Biology: Sporulation, Germination, and Corresponding Therapies for C. difficile Infection. Frontiers in Cellular and Infection Microbiology. 2018; 8: 29)

Lines 40-44: It is a bibliographic reference study

Response: Thank you for your suggestion. New bibliographic references have been added. (line 44)

Lines 44, 52, 54: Please correct the names of gram-negative and gram-positive bacteria. The entire manuscript needs correction.

Response: Thank you for your reminder. We correct them in the revised edition.

Lines 61-63: Please insert bibliographical references

Response: Thank you for your suggestion. New bibliographic references have been added. (line 61)

Lines 101-102: Who conducted this study?

Response: Thank you for your suggestion. New bibliographic references have been added. (line 102)

No information is mentioned about the spread of this method in clinical diagnostic laboratories, its costs, or its profitability.

Response: We appreciate the precise comments from the reviewer. We put the information in the Conclusion part as “Our DI method emerges as a solution that offers a rapid and easy-to-use alternative for identifying pathogens in blood cultures within just one hour, leading to a substantial time-saving of 5 hours compared to the STI technique commonly used in clinical microbiology laboratories. As previously mentioned, each hour gained in initiating appropriate antimicrobial therapy significantly enhances the chances of patient survival, making timely pathogen identification a critical factor in the management of sepsis [22]. The efficacy of DI method closely corresponds to that of the STI method, particularly in the case of Gram-negative bacteria. Despite molecular assays being the most efficient in pathogen detection in BCs, their limited usage is primarily attributed to their high expenses. Addition-ally, these tests' panels can only detect specific species [25–30]. Hence, our DI method proves to be more efficient than these molecular assays. The DI method involves utilizing BC pellets in conjunction with MALDI-TOF and Gram staining for the swift and precise diagnosis of bloodstream infections. By eliminating the need for incubation, this method facilitates the prompt identification of bacteria in samples without incurring additional costs, which is particularly crucial for urgent diagnostic procedures.” (page 18, line 127-141)

From my point of view, the number of samples processed in a year is relatively low, especially since it is a hospital. What is the capacity of the hospital under study?

Response: We appreciate the meaningful comments from the reviewer. To be clarified, we rewrite the sentence as “During January 2022 to December 2022, we collected 369 positive BCs from patients who were admitted to the hospital in a random manner” (page 3, lines 118-119)

Round 2

Reviewer 1 Report

Comments and Suggestions for Authors

Abstract

The authors should add a context/state of the art before the objectives. Results should be shortened. The need to develop new method should be added, highlighted.

Response: We appreciate the constructive comment from the reviewer. We rewrite the Abstract as “Timely pathogen identification in bloodstream infections is crucial for patient care. Comparison made between positive blood culture (BC) pellets from serum separator tubes using direct identification (DI) method and colonies on agar plates from short-term incubation (STI) method with matrix-assisted laser desorption/ionization Biotyper for the evaluation of 354 monomicrobial BCs.” (page 1, lines 16-20)

à short version but acceptable

à lesser instead of poor line 25

Introduction

The authors should highlight the need of new method and the fact that they have developed the “in-house” DI.

Response: We appreciate the suggestion from the reviewer. The need of new method and the fact that they have developed the “in-house” DI was put in the introduction part. (page 2-3, line 87-114)

à please indicate DI abbrevation line 91

à the introduction should be completed by adding sentence explaining their objectives including the development of a DI method.

Materials and methods

Why the authors performed only one sample per patient?

Response: A patient may acquire a recurrent bacterial infection within a brief period. Consequently, a singular sample is obtained from each patient. As a result, only one sample per patient was included in the study to avoid repetition. (page 3, line 127)

Why they incubated all specimen at 37°C and in a 5% CO2 environment since bacteria do not have the same optimal growth temperature?

Response: The primary clinical goal is to rapidly and accurately identify pathogens. The conditions of 37°C and 5% CO2 generally achieve this for most cases. While not optimal for all bacteria, these conditions cover the growth needs of most clinically relevant pathogens and provide a standardized, practical, and efficient operational environment. To be clarified, we add a sentence “Additionally, CDC ANA media were incubated at 37°C for 5 days, under anaerobic conditions.” (page 3, line 134-135)

DI method: the authors must give the centrifuge speed in g and not in rpm. They must indicate the temperature at which they centrifuge.

Response: We appreciate the constructive comment from the reviewer. We rewrite the sentence as “these tubes were spun at room temperature at 1,720 g for a period of 10 mins.” (page 3, line 143) and “subjected to further centrifugation at room temperature at a speed of 9,178 g for 5 mins.” (page 3, line 147-148)

What is the gel layer? to which biological elements it corresponds?

Response: We used BD Vacutainer Blood Collection separator tubes in the DI method. BD Vacutainer® SST™ Tubes contain spray-coated silica and a polymer gel. Its main function is to separate the serum or plasma from the blood cells during centrifugation, maintaining the purity and stability of the sample, thus facilitating accurate clinical tests and analyses.

à informations on biological elements present in the gel layer should be added into the text

The figure 1 is not in adequation with the information within the text (temperature of incubation, lack of information about the results of the second centrifuge in di method).

Response: We appreciate the suggestion from the reviewer. It is recommended to retain the existing figure 1 for a succinct depiction. Elaborate information has been included in the textual content. However, the centrifuge speed is presented in g instead of in rpm.

In case of already published methods for STI &/or DI, references should be added with information about optimizations or changes pointed out.  

Response: We add the reference [10,11] for STI method. (page 4, line 162) DI was performed using an in-house pretreatment procedure.

High speed centrifugation: at which speed? for how long? at which temperature??

Response: The centrifuge speed, timing and temperature were prescribed at DI method. (page 3, line 142-148) To be clarified, we rewrite the sentence as “Positive BC bottles were subjected to pretreatment, including cell removal and high-speed centrifugation, which was mentioned before to obtain the bacterial pellets.” (page 5, line 181-182)

à authors must change the sentence since 10000g is a not an high-speed centrifugation.

Results

à line 206 : the authors should indicate towards what DI is compared in this sentence.

How the authors explain the absence of a peak while using the DI method which seems to represent one drawback of their method (at least 1/3 of results)

Response: We appreciate the constructive comment from the reviewer. The absence of peaks when using the direct blood bottle identification method almost always occurs with Gram-positive bacteria and yeasts. The possible reason is the cell wall structure. Gram-positive bacteria have a thicker cell wall mainly composed of peptidoglycan, which makes protein extraction and analysis difficult. In MALDI-TOF analysis, the mass spectra of proteins are crucial for identifying bacteria; thus, the thick cell wall affects the quality and clarity of the mass spectra. Effective cell disruption is key to successfully extracting internal proteins and generating high-quality mass spectra. Existing cell disruption techniques, such as physical disruption (e.g., sonication) or chemical disruption (e.g., enzyme treatment), may be insufficient for Gram-positive bacteria and yeasts, leading to low protein extraction efficiency. Improving protein extraction techniques should address these issues. To highlight the issue, we rewrite the phrase It is worth noting that the manufacturer recommended the addition of 1 μL of formic acid post the deposition of microbial material on the MS target for the identification of Gram-positive bacteria and yeasts, which could have potentially impacted the quality of identification in this particular study. The utilization of formic acid on the MS target was executed through both DI and STI methodologies within the scope of this investigation. An alternative strategy improving protein extraction techniques should address these issues.” (page 17, line 111-117)

The authors should explain the difference scores and their relevance in the method used.

Response: MALDI-TOF MS results were interpreted according to the manufacturer's technical specifications as follows: score < 1.7 indicates no reliable identification, a score between 1.7 and 1.999 indicates identification to the genus level, and a score ≥ 2 indicates identification to the species level. (page 4-5, line 177-180)

The tables 2 & 3 must be redrawn to fit on only one page each, in a portrait mode. They should contain the same column in order to compare the results obtained.

Response: We appreciate the suggestion from the reviewer. We revised the Table 2 & 3 contain the same column to compare the results obtained.

à the tables 2 and 3 must be redrawn to fit within one page and the bacteria should be listed alphabetically

The table 4 should be replaced by a histogram to allow better comparisons between both DI & STI methods.

Response: Thank you for your valuable feedback. We propose retaining the present format of Table 4 owing to the presence of 11 strains and the diverse samples acquired for each strain.

à Histogram will allow to include p information and to understand the comparison used to determine this value.

Discussion

L32-40: the lines should be added in the abstract part

Response: We appreciate the suggestion from the reviewer. We rewrite the Abstract as “Timely pathogen identification in bloodstream infections is crucial for patient care. Comparison made between positive blood culture (BC) pellets from serum separator tubes using direct identification (DI) method and colonies on agar plates from short-term incubation (STI) method with matrix-assisted laser desorption/ionization Biotyper for the evaluation of 354 monomicrobial BCs.” (page 1, lines 16-20)

L52: DI was able to detect gram+ bacteria (table 1). Could you detail this point?

Response: We appreciate the constructive comment from the reviewer. To be clarified, we rewrite the phrase as “Our findings indicated that both DI and STI approaches demonstrated comparable levels of identification accuracy across a range of bacterial strains, except for Gram-positive bacteria, such as S. aureus and coagulase-negative staphylococci. In particular, 78 (46.7%) isolates out of 167 samples of Gram-positive bacteria could not be recognized utilizing the DI method. On the other hand, 18 (11.8%) isolates out of 152 samples of Gram-negative bacteria remained unidentified when applying the DI technique (Table 1).” (page 16, line 51-57)

à the sentence utilizing DI method should be rewritten into a positive manner, explaining that 53.3% of Gram + bacteria have been recognized. They should explain the difference between DI & STI and reasons of difference of recognition between both methods. 

L78-l80: how could the authors overcome the problem encountered with gram-positive bacteria & yeasts?

Response: We appreciate the suggestion from the reviewer. Effective cell disruption is key to successfully extracting internal proteins and generating high-quality mass spectra. Existing cell disruption techniques, such as physical disruption (e.g., sonication) or chemical disruption (e.g., enzyme treatment), may be insufficient for Gram-positive bacteria and yeasts, leading to low protein extraction efficiency. Improving protein extraction techniques should address these issues. We rewrite the phrase asIt is worth noting that the manufacturer recommended the addition of 1 μL of formic acid post the deposition of microbial material on the MS target for the identification of Gram-positive bacteria and yeasts, which could have potentially impacted the quality of identification in this particular study. The utilization of formic acid on the MS target was executed through both DI and STI methodologies within the scope of this investigation. An alternative strategy improving protein extraction techniques should address these issues.” (page 17, line 111-117)

à the authors should add information about alternative strategies

L103-109: the sentences should be placed as a conclusion part

Response: Thank you for your valuable feedback. The sentence has been placed as a conclusion part. (page 18, line 127-132)

à conclusion : the first sentence should be changed to explain the results obtained with their in house DI. The second sentence is not necessary within the conclusion part. Moreover, conclusion should give informations about the problem encountered with their DI method to detect Gram + bacteria.

L120-121: why the authors and how they evaluated DI method for anti-microbial susceptibility?

Response: We appreciate the constructive comment from the reviewer. The phrase “While the DI method proved to be valuable for pathogen identification, it did not show efficacy in antimicrobial susceptibility testing. The knowledge of local antibiograms of microorganisms, especially those with intrinsic resistance such as Pseudomonas sp. and Stenotrophomonas sp., can significantly aid in treatment management. The swift identification of pathogens can assist clinicians in choosing the most appropriate antimicrobial therapy [23,24]; nonetheless, further investigations are necessary to explore the impact of the DI method on the clinical outcomes of patients.” was put in the Discussion part. (page 17, line 119-125)

Conclusion

L132: does the duration of 5-hour valuable in every case or only for shorter ones? this point should be quantified in result part and discussed afterwards.

Response: We appreciate the precise comments from the reviewer. To be clarified, we rewrite the sentence “The mean time interval between bacterial growth detection as reported by the BC incubation system and the results obtained through the STI method was 6±1.5 hours in general. Conversely, the average duration associated with the DI method was 1±0.4 hours.” in result part. (page5, line 203-206) The key benefit of employing the DI technique is its ability to significantly diminish the time required for bacterial identification by 5 hours in comparison to the STI method. This time-saving aspect is particularly critical in scenarios necessitating prompt identification and commencement of suitable antibiotic treatment to influence patient outcomes, especially in instances of severe sepsis or other life-threatening infections.

à the sentence should be changed, as already asked before to understand the methods compared. moreover, the information should also be added into the conclusion.

Author Response

We have provided our point-by-point responses to the comments below. The changes in the revised edition are highlighted in red color. We hope you will find our revised manuscript suitable for publication in Diagnostics.

Reviewers' comments:

ROUND 2

Reviewer #1:

Abstract

à short version but acceptable

Response: Thanks.

à lesser instead of poor line 25

Response: We appreciate the constructive comment from the reviewer. We replace “poor” with “less”. (page 1, line 25)

Introduction

à please indicate DI abbreviation line 91

Response: We appreciate the suggestion from the reviewer. Change had been made accordingly. (page 2, line 91)

à the introduction should be completed by adding sentence explaining their objectives including the development of a DI method.

Response: We appreciate the constructive comment from the reviewer. We rewrite the sentence as “However, these kits entail numerous complex procedures, often leading to suboptimal identification result and high costs [15]. Hence, in-house protocols have been implemented. Consequently, the present research aims to develop a straightforward, practical, and cost-efficient approach for promptly identifying pathogens in positive BCs.” (page 3, line 107-111)

Materials and methods

à information on biological elements present in the gel layer should be added into the text

Response:  We appreciate the constructive comment from the reviewer. We add the phrase “The tubes contain spray-coated silica and a polymer gel. Its main function is to separate the serum or plasma from the blood cells during centrifugation, maintaining the purity and stability of the sample, thus facilitating accurate clinical tests and analyses.” (page 3, line 141-144)

à authors must change the sentence since 10000g is a not an high-speed centrifugation.

Response:  We appreciate the constructive comment from the reviewer. To be clarified, we rewrite the sentence as “Positive BC bottles were subjected to pretreatment, which was mentioned before to obtain the bacterial pellets.” (page 5, line 183-184)

Results

à line 206 : the authors should indicate towards what DI is compared in this sentence.

Response: We appreciate the constructive comment from the reviewer. To be clarified, we rewrite the sentence as “The period required for pathogen identification in positive BCs was notably reduced by 5 hours when employing our in-house DI method as opposed to the STI method.” (page 5, line 207-209)

à the tables 2 and 3 must be redrawn to fit within one page and the bacteria should be listed alphabetically

Response: Thank you for your valuable feedback. We try our best to redraw the Table 2 and 3. The bacteria have been listed alphabetically.

à Histogram will allow to include p information and to understand the comparison used to determine this value.

Response: Thank you for your valuable feedback. We draw the new Figure 3 to replace the Table 4 in the revised edition.

Discussion

à the sentence utilizing DI method should be rewritten into a positive manner, explaining that 53.3% of Gram + bacteria have been recognized. They should explain the difference between DI & STI and reasons of difference of recognition between both methods.

Response: We appreciate the constructive comment from the reviewer. To be clarified, we rewrite the phrase as “Our findings indicated that both DI and STI approaches demonstrated comparable levels of identification accuracy across a range of bacterial strains, except for Gram-positive bacteria, such as S. aureus and coagulase-negative staphylococci. In particular, 87 (52.1%) isolates out of 167 samples of Gram-positive bacteria could be recognized utilizing the DI method. On the other hand, 134 (88.2%) isolates out of 152 samples of Gram-negative bacteria were identified when applying the DI method (Table 1). Adequate cell disruption plays a crucial role in the efficient extraction of intracellular proteins and the production of high-quality mass spectra. The cell disruption methods employed in our in-house DI approach may prove inadequate for Gram-positive bacteria, resulting in suboptimal protein extraction efficiency and consequently lower identification performance.” (page 13, line 55-62)

à the authors should add information about alternative strategies

Response: We appreciate the suggestion from the reviewer. We rewrite the phrase as “An alternative strategy such as physical disruption (e.g., sonication) or chemical disruption (e.g., enzyme treatment) improving protein extraction techniques should address these issues.” (page 14, line 121-123)

Conclusion

à conclusion : the first sentence should be changed to explain the results obtained with their in-house DI. The second sentence is not necessary within the conclusion part. Moreover, conclusion should give information about the problem encountered with their DI method to detect Gram + bacteria.

à the sentence should be changed, as already asked before to understand the methods compared. moreover, the information should also be added into the conclusion.

Response: We appreciate the precise comments from the reviewer. We rewrite the first sentence as “Our in-house DI method offers a rapid and easy-to-use alternative for identifying pathogens in BCs within just one hour, leading to a substantial time-saving of 5 hours compared to the STI technique commonly used in clinical microbiology laboratories.” (page 16, line 137-139) The second sentence has been removed in the revised edition.

To be clarified, we rewrite the sentence “However, the cell disruption techniques utilized in our in-house DI strategy may not be sufficient for Gram-positive bacteria and yeasts, resulting in suboptimal extraction of proteins and subsequently lower identification accuracy.” to address the problem encountered with our DI method. (page 16, line 141-143)

Round 3

Reviewer 1 Report

Comments and Suggestions for Authors

major revisions

- why the authors separated anaerobic bacteria in table 2 ?

- the paragraph corresponding to the former table 4 / new figure 3 should be adapted / reformulated according to the new figure.

minor revisions

- term DI should be included into the sentence added in the introduction

- legend of figure 2 should be on the same line.

- tables 2&3 should be in a portrait format instead of a paysage format

- figure 3 : all p informations should be erased, only stars indicating significant difference should stayed at the corresponding place; what is CoNS ? numbers at the top of histograms could also be erased; abreviation GPB should be indicated to replace Gram-positive bacteria; title of the figure must be erased; n number should be placed under the corresponding term

Author Response

We have provided our point-by-point responses to the comments below. The changes in the revised edition are highlighted in red color. We hope you will find our revised manuscript suitable for publication in Diagnostics.

Reviewers' comments:

ROUND 3

Reviewer #1:

- why the authors separated anaerobic bacteria in table 2?

Response: We appreciate the suggestion from the reviewer. We delete the mark for anaerobes in the Table 2 & 3.

- the paragraph corresponding to the former table 4 / new figure 3 should be adapted / reformulated according to the new figure.

Response: We appreciate the constructive comment from the reviewer. The table 4 is replaced as Figure 3 in the revised edition.

minor revisions

- term DI should be included into the sentence added in the introduction

Response: We appreciate the suggestion from the reviewer. The term DI is included in the introduction. (Line 91, Page 2)

- legend of figure 2 should be on the same line.

Response: Thanks. The legend of figure 2 is on the same line. (Line 221-223, Page 6)

- tables 2&3 should be in a portrait format instead of a paysage format

Response:  We appreciate the constructive comment from the reviewer. The new table 2&3 are in the portrait format.

- figure 3: all p information should be erased, only stars indicating significant difference should stayed at the corresponding place; what is CoNS? numbers at the top of histograms could also be erased; abbreviation GPB should be indicated to replace Gram-positive bacteria; title of the figure must be erased; n number should be placed under the corresponding term.

Response: We appreciate the constructive comment from the reviewer. We revise the Figure 3 as reviewer’s suggestion.
